# The Gut Microbiota-Derived Immune Response in Chronic Liver Disease

**DOI:** 10.3390/ijms22158309

**Published:** 2021-08-02

**Authors:** Sung-Min Won, Eunju Park, Jin-Ju Jeong, Raja Ganesan, Haripriya Gupta, Yoseph Asmelash Gebru, SatyaPriya Sharma, Dong-Joon Kim, Ki-Tae Suk

**Affiliations:** Institute for Liver and Digestive Diseases, Hallym University College of Medicine, Chuncheon 24253, Korea; lionbanana@hallym.ac.kr or lionbanana87@gmail.com (S.-M.W.); epark312@hallym.ac.kr (E.P.); jj_jeong@hallym.ac.kr (J.-J.J.); RG@hallym.ac.kr (R.G.); phr.haripriya13@gmail.com (H.G.); yagebru@gmail.com (Y.A.G.); satyapriya83@gmail.com (S.S.); djkim@hallym.ac.kr (D.-J.K.)

**Keywords:** chronic liver disease, gut microbiome, immune response, gut–liver axis

## Abstract

In chronic liver disease, the causative factor is important; however, recently, the intestinal microbiome has been associated with the progression of chronic liver disease and the occurrence of side effects. The immune system is affected by the metabolites of the microbiome, and diet is the primary regulator of the microbiota composition and function in the gut–liver axis. These metabolites can be used as therapeutic material, and postbiotics, in the future, can increase or decrease human immunity by modulating inflammation and immune reactions. Therefore, the excessive intake of nutrients and the lack of nutrition have important effects on immunity and inflammation. Evidence has been published indicating that microbiome-induced chronic inflammation and the consequent immune dysregulation affect the development of chronic liver disease. In this research paper, we discuss the overall trend of microbiome-derived substances related to immunity and the future research directions.

## 1. Introduction

An enormous number of microorganisms organize symbiotic complexes with mammalian organisms [1]. The complex and sophisticated interrelationships between hosts and microbiota are drawing attention due to their effect on human physiology and disease susceptibilities [2]. The gut microbiota, which makes up a huge part of the microbial complex ecosystem, is made up of 100 trillion bacteria of diverse taxonomy, comprising ten-times more cells than human cells in the human body [3]. 

The gut microbiota provides a variety of beneficial bacterial products through metabolic activity, maximizes the efficiency of the host’s energy harvest, and promotes maturation of the intestinal immune system [4,5]. The homeostasis of such gut microbiota is strictly regulated by diverse factors and the mucosal immune system [6]. However, genetic factors in the host, the abuse of antibiotics, and changes in diet and lifestyle can alter the gut microbiota [7]. 

The host immune system and microbiota are deeply interrelated. Dysbiosis and disruption of the microbiota homeostasis are major factors causing chronic inflammatory and metabolic disorders [8,9]. Recently, research has found that a wide range of liver diseases are closely related to dysbiosis and intestinal microbial dysfunction [10,11]. The gut microbiota and liver have a bidirectional relationship, and, based on this relationship, bacterial products and metabolites from intestinal microbes can pass through the intestinal barrier and reach the liver through the portal circulatory system and can contribute to liver disease through several mechanisms [12,13]. 

In addition, the inflammatory response derived from the gut microbiota has been linked to several chronic liver diseases, particularly nonalcoholic fatty liver disease (NAFLD), alcohol-related liver disease (ALD), hepatocellular carcinoma (HCC), and cirrhosis. Treatments for various chronic liver diseases by targeting gut–liver axis damage are being sought through various interventions. In addition to antibiotics, methods, such as probiotics, prebiotics, synbiotics, bacterial metabolites, and fecal microbial transplantation are being studied.

The aim of this review is to identify the link between the immune response by the gut microbiota and chronic liver disease and to explain the effect of the gut microbiota on liver diseases. We will focus on clinical data and interventions for each pathology while noting the role of the gut–liver axis.

## 2. Chronic Liver Disease

Chronic liver disease is a continuous and gradual process in which the destruction of liver structures and the formation of regeneration nodules occur for more than 6 months [14]. The etiology of chronic liver disease is extensive and includes alcohol abuse, autoimmune diseases, toxins, and environmental and genetic factors. Chronic liver disease is a frequent and common clinical condition, and its prevalence has increased in recent years [15]. Chronic liver disease encompasses several conditions, including ALD, NAFLD, chronic viral hepatitis, and cirrhosis.

Chronic liver disease usually progresses to fibrosis. Liver fibrosis varies in the rate of progression depending on the etiology, environment, and genetic factors [16]. The onset of the common mechanism of hepatic fibrosis occurs in response to chronic liver injury, at which point inflammatory lymphocytes enter the hepatic parenchyma and some hepatocytes undergo apoptosis. After that, Kupffer cells are activated, and hepatic stellate cells are activated into proliferative fibrogenic myofibroblasts, secreting and accumulating extracellular matrix proteins, including collagen [17,18]. 

Disruption of the balance between the deposition and decomposition of extracellular matrix proteins leads to structural distortion of the liver tissue by the formation of fibrous scars, which leads to cirrhosis [19,20]. Additionally, advanced liver fibrosis, unlike common liver fibrosis, which is considered a local reaction, is considered an irreversible condition and is a major risk factor for HCC, and, in severe cases, a liver transplant is required for treatment [19].

After fibrosis, cirrhosis—the final stage of chronic liver disease—occurs and has a variety of causes. In developed countries, viral hepatitis C, ALD, and NAFLD are the primary causes, and, in developing countries, viral hepatitis B and viral hepatitis C have been reported as the primary causes [21]. Previously, liver damage from multiple causes was considered permanent and irreversible. Therefore, avoiding the cause of the injury, early treatment and managing complications were the treatment goals. However, recent clinical and animal studies have reported evidence that liver cirrhosis may be reversible, and cases of the regression of cirrhosis have been reported leading to active research on therapeutic agents [22].

## 3. Gut–Liver Axis and Immune Response

To maintain a close relationship between the gut microbiome and the host, the immune response essentially acts as a lubricant. This immune system influences the composition of the gut microbiota, while the regulation and maturation of immunity are influenced by the gut microbiota [7,23]. The specialized and sophisticated interactions of the immune system are evident in the mucosal immune system of the intestine [24]. The mucosal immune system is an interrelated system that protects the host by responding to external pathogenic attacks but preserves the beneficial microbiota, allowing them to thrive. 

Gut-associated lymphoid tissue (GALT), a major component of mucosal-associated lymphoid tissue, is made up of Peyer’s patches, congenital lymphocytes, and T and B cells, affecting the strength of the entire immune system and playing a key role in systemic and local immune responses [25]. Imbalances and dysregulation of the immune system in the gut and liver are associated with the onset and progression of intestinal and liver disease [26]. 

While the mucosal surface of the intestinal barrier serves as a primary barrier, mucus protects the basal epithelium, induces immunomodulatory signals, and maintains and enhances homeostasis. In the porous mucosal layer present in the small intestine, MUC2 mucin is directly absorbed by dendritic cells, imprinting anti-inflammatory properties on the dendritic cells [27]. 

These actions inhibit the expression of inflammatory cytokines by inhibiting gene signaling through nuclear factor-κB. The induction of regulatory signals in these dendritic cells of MUC2 limits and modulates the immunogenicity of gut antigens. The dendritic cells then migrate to the mesenteric lymph nodes and present antigens that stimulate Treg cells and effector T cells. These cells deliver regulatory cytokines, such as TGF-β, IL-10, and IL-35, throughout the body and carry out immune responses, while safeguarding the balance of the gut and immunity [6].

Maintaining an appropriate balance between beneficial and harmful bacteria is called “eubiosis”, and this condition is important for maintaining immune homeostasis. This balanced state is disrupted by various factors, such as an unbalanced diet and abuse of antibiotics, which is called “dysbiosis”. The outbreak of dysbiosis increases pathogenicity and can lead to several diseases, such as metabolic disorders [28]. In addition, disruption of the gut microbiota balance causes damage to the mucosal barrier and allows bacteria and bacteria derived products to enter the peripheral circulatory system [29]. Toll-like receptor (TLR) signaling is subsequently activated and increases the release of inflammatory cytokines, which can lead to systemic inflammation [30]. 

The immune response effects of the gut microbiota also apply to the liver. The supply of large amounts of blood circulating from the intestine to the liver enables the movement of endotoxins and bacteria-derived products, and the liver relies on the innate immune system to defend against it. Microbial-associated molecule patterns, such as LPS, lipoteichoic acid, peptidoglycan, and lipoproteins that are released into the portal vein, are detected by immune cells expressing pattern recognition receptors and trigger an activation [31]. 

The liver contains a huge number of innate immune cells, such as natural killer cells, natural killer T cells, macrophages, and γδ T cells [32]. It has been reported that substances derived from the gut microbiota affect the maturation of hepatic natural killer (NK) cells and the maintenance of the homeostasis of hepatic interleukin (IL)-17A-producing γδ T cells [33,34]. Throughout this process, evidence of tumor growth inhibition or acceleration of nonalcoholic liver disease progression has been observed. In addition, the gut microbiota affects the liver by releasing microbial-derived molecular patterns, such as lipopolysaccharide and endotoxins, into the portal circulation. 

However, the liver acts as a firewall by filtering bacteria or derived substances released into the hepatic portal vein [35]. When the microbial-derived molecular patterns reach the liver, receptors, such as TLR4, activate Kupffer cells, hepatic stellate cells, and hepatic sinusoidal endothelial cells and can induce an inflammatory response [36,37]. The increase in microbial-derived molecular patterns by dysbiosis affects the liver’s immune environment through the regulation of inflammatory cytokines. 

Activation of lipopolysaccharide (LPS)-TLR4 induces the secretion of proinflammatory cytokines, such as tumor necrosis factor (TNF)-α and IL-6 in Kupffer cells, and excessive cytokine secretion with continuous LPS accumulation may act as a pathological mediator of inflammation-related HCC [38]. Eventually, sustained bacterial translocation or an increase in microbial-derived molecular patterns, triggered by intestinal dysbiosis can lead to excessive immune responses that threaten the health of the host. Recent studies reported accumulating evidence that dysbiosis and specific microbial taxa are associated with chronic liver disease, with immune-related responses, such as the activation of the innate and adaptive immune responses, and with the production or suppression of inflammatory cytokines [39,40] (Figure 1).

Bile acids (BAs) are synthesized from cholesterol in hepatocytes, bound to glycine or taurine, and released into the bile ducts. BAs promotes the emulsification and absorption of fats, cholesterol, and fat-soluble vitamins in the small intestine, after which 95% of the BAs are reabsorbed from the ileum back to the liver [41]. The remaining 5% are reprocessed by the gut microbiota and reach the liver through the portal vein in the form of secondary bile acids. 

This enterohepatic circulatory system plays an important role in maintaining homeostasis as part of the gut–liver axis. BA directly controls the gut microbiota and binds with FXR to induce the production of antimicrobial peptides, such as angogenin1 [42]. Through this, it suppresses intestinal microbial overgrowth and intestinal barrier dysfunction. However, dysbiosis disrupts primary and secondary bile acid circulation and enterohepatic circulation balance and triggers a series of host immune responses, contributing to the progression of liver disease [43].

## 4. Gut Barrier Dysfunction

The intestinal barrier acts as a barrier against infectious agents, such as toxins and bacteria, entering the circulation [44]. The intestinal epithelium forms a tight physical junction, allowing for the selective absorption of nutrients. The intestinal barrier is constructed by the binding of enterocytes by transmembrane proteins consisting of tight junctions, adherens junctions, and desmosomes [44]. In addition, the intestinal barrier is reinforced by commensal bacteria, a layer of mucins, and various immunoglobulins. However, various factors alter or disrupt the function of the intestinal barrier. 

Dysbiosis causes changes in certain gut microbiota taxa, leading to the inhibition of mucus production, degradation of the mucus layer, and alteration of tight junctions [44]. Endotoxins from Gram-negative bacteria increase the expression of TLR4 and the permeability of tight junctions [45]. Increased intestinal permeability generates endotoxin and bacterial translocation and enables entry into the liver through portal circulation [46]. This results in a systemic inflammatory response and liver damage, which contributes to chronic liver disease. 

Intestinal barrier disorders caused by dysbiosis can be accelerated by factors, such as diet and alcohol. Increased intestinal permeability was observed in mouse models fed a high-fat or choline-deficient diet and in patients with NAFLD [47,48]. An unbalanced diet, such as a high fat diet, causes an abnormal composition of the gut microbiota. The altered gut microbiota composition leads to increased bacterial permeability, reduced thickness of the mucosal layer, redistribution of tight junction proteins in the epithelial barrier, and low-grade intestinal inflammation [49]. 

In the results of fecal microbial transplantation from a high-fat diet mouse model to a normal diet mouse model, damage to the intestinal barrier was confirmed, indicating that the altered composition of the gut microbiome rather than the diet itself is a major factor in intestinal barrier damage [50]. Increased intestinal permeability is a well-known feature in alcoholics, chronic alcohol abuse, alcoholic mouse models, and patients with alcoholic liver disease at the cirrhosis stage [51]. In addition to the toxic effects of alcohol and its derived metabolites on intestinal epithelial cells, there is sufficient evidence that dysbiosis due to alcohol contributes to intestinal barrier dysfunction and bacterial translocation [52]. 

Increased serum endotoxin levels and bacterial DNA have been identified in patients and mouse models after acute or chronic alcohol abuse [53]. These results occur because alcohol damages certain components of the intestinal barrier. In the alcohol mouse model, the inhibition of intestinal regenerating islet-derived protein 3-β (Reg3b) and 3-γ (Reg3g) was confirmed upon the administration of alcohol [47]. This can lead to phenomena, such as intestinal bacterial overgrowth, bacterial translocation, and the exacerbation of liver inflammation [54]. In summary, intestinal barrier dysfunction and translocation of bacteria and, thus, the products through them can be a significant cause of chronic liver disease and related complications [55]. 

Currently, although intestinal barrier function cannot be concluded to be significantly correlated with endotoxemia, increased intestinal permeability is, at least in part, implicated in the pathophysiology of several liver diseases [56]. The ‘leaky gut’ hypothesis still links microbial products in the gut with the pathogenesis and progression of NAFLD and ALD and has long been considered one of the major contributors. Compared with healthy controls, patients with NAFLD show increased intestinal permeability and tight junctions, and chronic alcohol abuse contributes to the disruption of the intestinal barrier, which is critical for the development and progression of ALD, thereby, supporting this hypothesis.

## 5. Immune Response Associated with Liver Disease

### 5.1. Nonalcoholic Fatty Liver Disease and the Immune Response

NAFLD encompasses a wide range of liver diseases, from simple hepatic steatosis to nonalcoholic steatohepatitis (NASH), and cirrhosis [57]. NAFLD is also closely related to metabolic diseases, such as obesity, and shares a common cause and mechanism. However, what makes NAFLD so distinct from obesity is the difference in a process called lipotoxicity. In the process of liver lipid overload, the way liver cells deal with this is either steatosis adaptation or the induction of cell death by molecular mechanisms [58]. 

Stress signals released from hepatocytes due to cell death trigger the activation of inflammatory pathways and, over time, lead to abnormal wound repair processes, such as chronic injury and liver fibrosis [59]. In this way, NAFLD can deepen and progress to NASH and fibrosis. Recently, the influence of the gut microbiome according to the gut–liver axis relationship as a factor in the novel NAFLD pathogenesis has been attracting attention.

Studies have shown that dysbiosis contributes to the development of NAFLD; however, no clear causal relationship has yet been established. However, several preclinical studies and clinical studies have reported that metabolites produced by specific gut microbiota are associated with the expression of simple steatosis and NASH [60]. Dynamic changes in the gut–liver axis, such as microbial-derived metabolites and bacterial infiltration, are the result of alterations in intestinal permeability associated with the development of NAFLD [61]. 

Dysbiosis factors, such as the overgrowth of gut microbiota and changes in the gut microbiota composition, were identified in NAFLD patients compared to healthy controls, supporting a correlation with the intestinal permeability [62,63]. Reported features of patients with NAFLD include increased intestinal permeability, overgrowth of small intestinal bacteria, and increased serum endotoxins [64,65,66]. Dysbiosis causes disruption of the intestinal barrier and translocated microbial-derived products, such as LPS, triggering TLR4 activation and an inflammatory cascade [67,68]. 

In NAFLD, choline metabolites are one of the important factors in the pathogenesis and progression of the disease. Choline is an essential nutrient that is important for maintaining a healthy metabolism and plays a key role in liver functions, brain development, and nerve functions [69]. Choline plays a role in helping the liver to excrete particles of very-low density lipoproteins and prevents hepatic steatosis. These properties allow a choline-deficient diet to mimic nonalcoholic steatohepatitis in a mouse model [70]. Choline is converted to trimethylamine (TMA) by the intestinal microflora and then to trimethylamine N-oxide (TMAO), which can be transported to the liver. Increased systemic circulation of TMAO leads to hepatic steatosis, which leads to liver damage and is another cause of NAFLD [71].

In a NAFLD-induced TLR4 mutant mouse model fed a methionine/choline-deficient diet, liver damage and lipid accumulation were decreased compared to those in the control group [72]. In addition, due to depleting the liver Kupffer cells by the continuous injection of clodronate liposomes, the increase in TLR4 expression in the liver was prevented, and histological changes in fatty hepatitis were observed [72]. When Kupffer cells are activated by LPS, they secrete proinflammatory cytokines, such as IL-1β, IL-18, and IL-12, inducing the activation of innate immune cells [73]. 

In another study, a significant decrease in hepatic triglyceride accumulation was observed in TLR4 mutant mice fed fructose, and the hepatic lipid peroxidation and levels of TNF-α and MyD88 were significantly decreased [74]. TLR4, which is activated by LPS, regulates the expression of hepcidin, a key protein related to NAFLD, through MyD88 [75]. LPS/TLR4 signaling also plays a key role in the activation of fibrogenesis in hepatic stellate cells. The interaction between hepatic stellate cells and activated Kupffer cells accelerates fibrosis with increased transforming growth factor-β (TGF-β) [76] (Table 1). 

In another study, it was confirmed that nucleotide- binding oligomerization-domain protein-like receptor protein (NLRP) 6 and NLRP3 inflammasomes and IL-18 negatively regulate the progression of NAFLD or NASH through gut microbiota modulation [77]. Alterations in the gut microbiota composition in inflammasome-deficient mice were associated with the exacerbation of hepatic steatosis and inflammation [77,78]. Research found that microbial-derived products acting as agonists of TLR4 and TLR9 influx into the portal circulation, strongly inducing TNF-α expression and affecting the progression of NASH [79]. The distorted gut–liver axis interactions caused by defective NLRP3 and NLRP6 inflammasome sensing can influence the rate of progression of NAFLD and NASH, suggesting that the gut microbiota play a key role in systemic autoinflammation and pathogenesis.

### 5.2. Alcoholic Liver Disease and the Immune Response

The underlying cause of ALD is chronic alcohol abuse. This includes a variety of histological phenomena ranging from hepatic steatosis, in which fat accumulates in the liver, to hepatic inflammation in progressive ALD and to fibrosis/cirrhosis [82]. Almost all alcohol abusers develop hepatic steatosis, 10% to 35% of whom develop alcoholic hepatitis and 8% to 20% of whom develop cirrhosis [83]. 

Chronic inflammation is a key etiology of ALD, and it has been reported that ethanol itself enhances the ability of immune cells to respond to inflammatory stimuli. Alcohol exposure upregulates the expression of TLRs and enhances the signaling of NF-κB and Early Growth Response-1, induced proinflammatory cytokine production [84,85]. Additionally, it increases the proinflammatory activity of hepatic NKT cells and promotes the production of chemokines, such as IL-8 and MCP-1 [86,87].

In recent studies, attention has been focused on elucidating the relationship between the gut microbiota and ALD. How alcohol causes dysbiosis of the gut microbiota and affects the pathogenesis of ALD is being identified at the preclinical and experimental levels [88]. In this process, evidence is being gathered regarding the toxic components and pathogens involved in the pathogenesis of ALD [89]. However, the data are still insufficient at the clinical level. It is becoming increasingly clear that the inflammatory response triggered by dysbiosis is a key factor in the progression from alcoholic liver damage to alcoholic liver disease.

Several factors play a role in how the gut microbiota affects susceptibility to ALD. Major factors include acetaldehyde caused by the ethanol metabolism, increased intestinal inflammation due to gut dysbiosis, and changes in bile acids and metabolites [90,91,92,93]. Ethanol and its metabolites acetaldehyde and acetate are closely linked to liver damage [94]. Ethanol also damages the intestinal barrier, allowing for bacterial translocation and the entry of microbial products, and causes the weakening of tight junctions [95]. Ethanol down-regulates the expression of antimicrobial peptides in the intestine, weakening the inhibition of bacterial overgrowth, which also leads to a decrease in intestinal butyrate [96,97]. 

Since most of the intestinal venous blood is transferred into the portal vein, microorganisms and metabolites can reach the liver [88]. Pathogen molecule pattern recognition receptors, such as TLRs and NLRs, recognize LPS and bacterial DNA and induce the activation of Kupffer cells and invasive macrophages [98]. The inflammatory cytokines and chemokines produced by this process will affect the progression of the disease. Cytokines and chemokines, such as CC-chemokine ligand 2 (CCL2), IL-8, and IL-1b are representative of this process. 

The relationship between gut-microbiota-derived products and the etiology of alcoholic liver disease has been demonstrated in several studies. It was reported that the intestinal sterilization treatment of mice through antibiotic treatment reduces ALD by preventing alcohol-induced liver damage [99]. In addition, high levels of endotoxins in the plasma of alcoholics have been identified [100,101], and their presence was demonstrated to exacerbate the disease through elevated inflammatory cytokines, such as IL-6 and IL-8 [102]. 

The effects of LPS in alcoholic liver disease are primarily mediated by TLR4 in Kupffer cells. Kupffer cells are the primary cell type that responds to LPS, and they act similarly to the inflammatory response mechanism of nonalcoholic liver disease. Other cell targets in the liver of LPS include hepatic stellate cells and sinusoidal endothelial cells. In the results of LPS pretreatment on hepatic stellate cells, it was observed that the secretion of collagen and IL-6 was increased, and the damage caused by alcohol was exacerbated [103]. 

In addition, LPS triggers the release of cytokines and chemokines through stimulation of sinusoidal endothelial cells [104]. Another study suggested that patients with alcoholic hepatitis have increased fecal numbers of *Enterococcus faecalis*. As bacteriophages can specifically target cytolytic *E. faecalis*, this provides a method for precisely editing the intestinal microbiota [105].

The excessive intake of alcohol affects liver damage and lipid accumulation, as well as increases TLR4 expression. This potentially changes the sensitivity of the TLR4 signaling system in the liver. A study in which alcohol was fed to hepatitis C virus NS5A transgenic mice showed that activation of TLR4 in mouse hepatocytes increase the sensitivity to alcohol and LPS, leading to liver damage and liver tumor formation [106]. Accumulating evidence indicates that LPS plays an important role in liver damage; however, LPS alone cannot mimic alcohol steatosis or alcoholic hepatitis in disease models. 

When fed with alcohol, the effects of worsening liver damage progress synergistically. The synergistic mechanisms involved in alcohol and LPS include factors, such as sympathetic stimulation [107], NADPH oxidase-dependent reactive oxygen species (ROS) production [108], and nuclear factor-kB activation [109]. In addition to bacterial products such as LPS, the negative effects of mycobiome-derived products on alcoholic liver disease have recently been reported. 

In fecal samples from alcoholic hepatitis patients, an increase in the abundance of *C. albicans* and the related substance, the extent of cell elongation 1 (ECE1), was observed [110]. These mycobiome-derived candida lysin may affect the severity and mortality of alcoholic hepatitis patients and support the broad association of the microbiome with liver disease, highlighting the need for additional research (Table 2).

### 5.3. Liver Cirrhosis and the Immune Response

Cirrhosis is the final stage of chronic liver disease caused by various mechanisms leading to severe fibrosis and liver damage [113]. Cirrhosis usually consists of compensatory and decompensatory stages and is accompanied by various complications. It has recently been shown that the gut microbiota, including bacteria, fungi and viruses, changes during the onset and progression of cirrhosis [114]. Several factors, including excessive alcohol intake, diet, and liver disease, alter the gut microbiome, leading to changes in the gut–liver axis [115]. 

Cirrhosis development is further accelerated through changes in the gut–liver axis and systemic inflammatory conditions [116]. Complications, such as hepatic encephalopathy, and hepatitis symptoms, such as bacterial peritonitis, are strongly associated with gut–liver axis interaction disruption [116]. Gut–liver axis disruption is manifested through phenomena, such as decreased intestinal motility, increased intestinal permeability, and small intestinal bacterial overgrowth, resulting in increased bacterial translocation and portal circulatory influx of endotoxins [117]. 

Subsequently, this contributes to the activation of hepatic stellate cells through TLR4 in the liver and leads to the activation of inflammatory pathways and the progression of fibrosis, which acts as in the pathogenesis of cirrhosis [118]. Studies have confirmed bacterial translocation and the increased activity of inflammatory cytokines in the portal blood of patients with cirrhosis [119,120].

Small intestine bacterial overgrowth is closely related to systemic endotoxemia. In a clinical study of cirrhosis patients, intestinal bacterial overgrowth was observed in 59% of patients [121]. In addition, a cirrhosis mouse model with intestinal bacterial proliferation exhibited high bacterial translocation and slow intestinal transit compared to the control group [122]. Similarly, it was confirmed that the excessive proliferation of *Enterobacteriaceae* through dysbiosis affected the bacterial translocation and liver function in cirrhosis animal studies [123]. 

TLR4 and gut microbiota-derived LPS, previously mentioned in other liver diseases, contribute to the development of hepatic fibrosis [124]. Alcohol has been shown to increase the sensitivity to gut microbiota-derived endotoxins. In alcoholic steatohepatitis patients, T-cell mobilization in the liver and collagen accumulation by activation of hepatic stellate cells were confirmed [125]. A recent study confirmed that TLR4 signaling, activated in hepatic stellate cells rather than in Kupffer cells, was more important for the development and progression of hepatic fibrosis [126].

Symptoms, such as small intestinal bacterial overgrowth, endotoxin activation, and bacterial translocation, occurring in liver cirrhosis can improve the clinical condition of cirrhosis patients through gut–liver axis management. Such gut–liver axis management enables inhibition of the development of fibrosis and endotoxin activity and may be effective in preventing cirrhosis (Table 3).

### 5.4. Hepatocellular Carcinoma and the Immune Response

The pathogenesis of HCC is caused by a combination of various factors. Non-viral HCC may contribute to the pathogenesis of the disease through hepatic steatosis, oxidative stress, and endoplasmic reticulum stress, and dysbiosis and the resulting inflammation are attracting attention as additional factors [115]. Data from clinical and animal studies allow us to observe extreme changes in the composition of the gut microbiome in hosts with HCC. 

In clinical studies with HCC patients, an excessive increase in *E. coli* was confirmed [128]. In addition, animal models and human studies have confirmed the presence of *Helicobacter* spp. in liver tissue samples [129]. This has been suggested as a potential mechanism by which *Helicobacter* migrates to HCC tumor tissue via intestinal translocation to suppress anti-tumor immunity and cause the activation of NF-kB signaling to promote carcinogenesis [130]. However, strangely, *Helicobacter* was not found in patients with viral HCC.

Dysbiosis, which is associated with HCC, increases the bacterial translocation and circulation of carcinogens through the disruption of the intestinal barrier and contributes to the activation of several proinflammatory and oncogenic signaling pathways [131]. Additionally, the role of the gut microbiome in liver tumorigenesis has been clearly identified in animal studies. The attenuation of liver inflammation and HCC development was observed using wide-spectrum antibiotics in a mouse model. In a mouse model in which HCC was induced through carcinogens, the influx of LPS led to the activation of TLR4, followed by the activation of the NF-kB pathway in HSC, resulting in increased tumor cell proliferation [132].

As our understanding of the gut microbiome increases, various therapeutic approaches for HCC are being devised. Research on reducing the risk of developing HCC in NAFLD through synthetic bile acids [133] and on suppressing tumor size and growth by administering probiotics to HCC mouse models is actively being conducted. However, further clinical studies are needed to link and characterize the role of the gut microbiota with the pathogenesis of HCC (Table 4).

### 5.5. Other Liver Disease and the Immune Response

Hepatic encephalopathy is a cognitive impairment caused by serious liver disease, and it has been confirmed that there is a deep relationship with the gut microbiome and metabolites. Ammonia is a key factor in the onset of hepatic encephalopathy, and overgrowth of urease-activated bacteria is known to be the main cause of hyperammonemia [134]. Recently, however, the focus of ammonia-generating sources has changed from the large intestine to the small intestine and kidney. 

Other factors that aggravate and develop hepatic encephalopathy include systemic inflammation and endotoxemia caused by dysbiosis, and animal studies confirmed that these factors caused the aggravation of comas and cytotoxic edemas in a cirrhosis model [135]. In addition, a decrease in neuropsychological function was confirmed after hyperammonemia was induced in cirrhosis patients with inflammation and infection [136]. 

As a therapeutic approach for hepatic encephalopathy, modulation of the gut microbiota is attracting attention as an alternative treatment, and the effects of prebiotics, probiotics, and synbiotics are being verified. In addition, a therapeutic approach through the administration of rifaximin has been confirmed to be safe and to efficiently control the gut microbiome, and studies are being conducted [137] (Table 4).

Primary sclerosing cholangitis (PSC) is a chronic liver disease characterized by biliary inflammation and stenosis of the bile ducts [138]. Various animal studies over the past few years have provided supporting data for a causal relationship between the gut microbiota and PSCs. Dysbiosis, bacterial translocation by weakening the intestinal barrier, and the immune response associated with the pathogenesis of PSC are considered to be key factors [139]. In animal studies, transplantation of fecal microbiota from PSC patients into germ-free mice showed that the PSC phenotype was transferred and that this increased the susceptibility to hepatobiliary damage by diethyldithiocarbamate [140]. 

In addition, in the MDR2−/− mouse model mimicking human PSC, researchers were able to confirm dysbiosis and increased intestinal permeability [141]. The gut microbiota contributes to PSC pathogenesis through interventions in the synthesis and production of various metabolites, including bile acids, which influence disease pathogenesis as signaling molecules in the gut–liver axis [142]. Although the identification of a potential causal relationship between the gut microbiota and PSCs in animal studies has made significant progress, the validation and reproducibility in clinical studies is still minimal. 

In another study with two cohorts, the gut microbiomes of PSC patients revealed functional differences compared with those of the control group, including the microbial metabolism of essential nutrients [143]. Regarding epithelial barrier dysfunction, *K. pneumoniae* disrupts the epithelial barrier to initiate bacterial translocation and liver inflammatory responses in PSC patients. In this study, antibiotic treatment ameliorated the T helper 17 immune response induced by PSC-derived microbiota [140] (Table 4). 

Primary biliary cholangitis (PBC), previously known as primary biliary cirrhosis, is a chronic liver disease caused by immune cell activation and damage to the bile ducts. PBC, which had been considered a typical autoimmune disease, through gut–liver axis interactions, has led to the inclusion of the influence of the gut microbiota in a revised etiological understanding [144]. 

Microbial-associated molecular patterns by dysbiosis can lead to persistent inflammation in the bile duct and aggravate the disease. The presence of pyruvate dehydrogenase complex E2 subunit antimitochondrial antibodies (AMA) is a representative serological characteristic of PBC, which cross-reacts with proteins of strains, such as *E. coli* [144] and *Novosphingobium aromaticivorans* [145], resulting in immune attacks on biliary epithelial cells. 

In animal studies, PBC-mimicking liver disease occurred in mice when a specific mouse model was infected with *N. aromaticivoransm* [146]. This is meant as supporting evidence for the association of PBC with the gut microbiome (Table 4). Novel therapeutic approaches based on the microbiome appear to be necessary for PBC management, and additional clinical data and clinical studies are needed.

**Table 4 ijms-22-08309-t004:** Hepatocellular carcinoma and other liver disease studies related to the immune response.

Species	Study Type	Exposure	Main Results	Ref.
HCC				
*Helicobacter*-free C3H/HeN female mice	Animal	AFB1 and/or *H. hepaticus*	Intestinal colonization by *H. hepaticus* promoted aflatoxin and HCV transgene-induced HCC. *H. hepaticus* activated the nuclear factor-kappaB regulatory signaling pathway.	[130]
C3H/HeOuJ, C3H/HeJ, TLR2-deficient mice, TLR4-deficient mice, TNFR1-/IL-1R1-double deficient, and C57Bl/6 mice	Animal	Intraperitoneal injection of DEN or CCl_4_	Activation of gut microbiota and TLR4 contributes to the development of cancer in chronically damaged livers. Intestinal microbiota and TLR4 contribute to promotion of HCC, proliferation of cancer, expression of hepatomitogen epiregulin and prevention of apoptosis. In the late stages of liver cancer, limited enteric sterilization reduced hepatocellular carcinoma.	[132]
Controls (*n* = 15), HCC patients (*n* = 15)	Human		The presence of HCC was associated with an increased number of *E. coli* in the patient’s stool. Intestinal *E. coli* overgrowth contributes to the development of liver cancer.	[128]
Controls (*n* = 16), patients with primary liver carcinoma (*n* = 20)	Human		*Helicobacter* spp. DNA was found in liver cancer samples from patients with primary liver carcinoma. By bacterial translocation, *H. pylori* may be present in the liver of liver carcinoma patients and may be related to hepatic carcinogenesis.	[129]
Hepatic Encephalopathy				
Male Sprague-Dawley rat	Animal	Bile duct ligation or Sham operation or High protein/ammoniagenic diet injected with LPS (0.5 mg/kg)	After LPS injection, only the bile duct ligation group progressed to the pre-coma stage. TNF-α and IL-6 levels were significantly increased in LPS-treated animals. LPS injection in a cirrhosis model induces coma due to synergistic effects of hyperammonemia and inflammatory response. It also exacerbates cytotoxic edema.	[135]
Cirrhotic patients (*n* = 10)	Human	Oral administration of an amino acid solution mimicking hemoglobin composition	Hyperammonemia was similar before and after resolution of inflammation in patients. There was a significant decrease in the white blood cell count, nitrate/nitrite, IL-6, IL-1β, and TNF-α by infection treatment. Induced hyperammonemia significantly worsened neuropsychological test scores.	[136]
PSC and PBC				
Germ-free C57BL/6 male mice	Animal	PSC patients fecal sample inoculation	T helper 17 cell responses were shown in the livers of Gnotobiotic mice inoculated with PSC patient-derived microbiota and increased susceptibility to hepatobiliary injury. PSC-associated Klebsiella pneumoniae has an epithelial damaging effect and contributes to bacterial translocation and initiation of hepatic inflammatory responses.	[140]
Male Mdr2−/−, Mdr2−/− crossed with hepatocyte-specific deletion of caspase-8 (Mdr2−/−/ casp8∆hepa) and wild-type (Wt) control mice	Animal	Administration of pan-caspase inhibitor (iDn-7314)	Abnormalities in the gut microbiome in Mdr2−/− mice caused intestinal barrier dysfunction and increased bacterial translocation, which amplifies the hepatic nlrP3-mediated innate immune response. Transfer of the Mdr2−/− microbiota to healthy wildtype control mice induced significant liver damage in recipient mice. MDr2-associated cholestasis causes intestinal bacterial imbalance. Translocation of endotoxin into the portal vein and subsequent nlrP3 inflammasome activation contributes to higher liver damage.	[141]
C57BL/6 and A/J mice onto the NOD background	Animal	Infection by intravenous injection of *N. aromaticivorans*	*N. aromaticivorans* infection induced liver inflammation and PBC.	[146]

AFB1, Aflatoxin B1; CCl_4_, carbon tetrachloride; DEN, Diethylnitrosamine; HCC, Hepatocellular Carcinoma; IL, interleukin; LPS, Lipopolysaccharide; PBC, Primary biliary cholangitis; PSC, Primary sclerosing cholangitis; and TNF-α, Tumor necrosis factor-α.

## 6. Conclusions and Future Directions

As we have described, chronic liver disease is associated with the microbiome, and one of the main mechanisms is immunomodulation. To date, the role of immunity in chronic liver disease has been initially studied with promising results. A new concept of immune regulation by microbiota has emerged and been demonstrated with firm evidence. Immunotherapy using the microbiome is being attempted through clinical studies in various fields, and extensive research results are expected in the future. As the microbiome is affected by several factors, it is currently being considered regarding adjuvant therapy for immunotherapy; however, microbiome treatments will likely be used as a primary treatment for individualized medicine in the future.

## Figures and Tables

**Figure 1 ijms-22-08309-f001:**
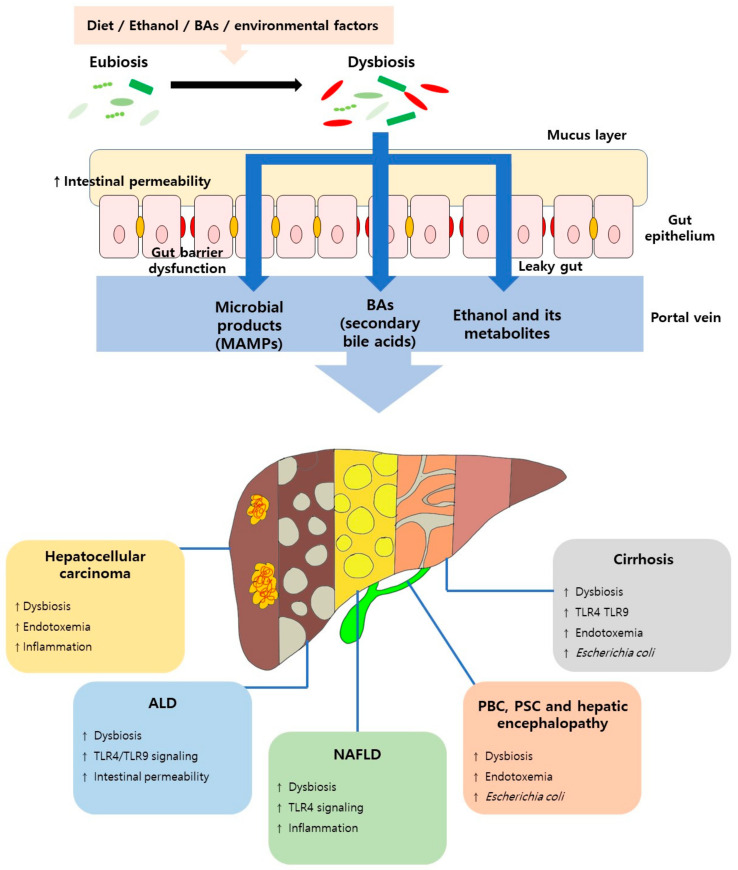
The role of the gut microbiota in liver disease. ALD, alcoholic liver disease; BAs, Bile acids; MAMP, Microbial-associated molecular patterns; NAFLD, nonalcoholic fatty liver disease; PBC, primary biliary cholangitis; and PSC, primary sclerosing cholangitis.

**Table 1 ijms-22-08309-t001:** Nonalcoholic fatty liver disease studies related to the immune response.

Scheme	Study Type	Exposure	Main Results	Ref.
Germ-free C57BL/6J female mice	Animal	Western-style diet and high-fructose diet	Although intestinal barrier damage was observed in the germ-free mouse group, hepatic steatosis did not occur due to the absence of aseptically induced LPS translocation. Required for commensal bacteria in the gut microbiota to induce hepatic steatosis by factors, such as diet	[80]
Germ-free C57BL/6J male mice	Animal	Normal chow diet and FMT in genetically obese human donor	The gut microbiota of genetically obese humans influences the hepatic transcriptional profile of lipid metabolism such as PPAR α in mice, promoting the pathogenesis of hepatic steatosis. High serum LPS levels in the obese group can suppress the expression of PPAR α.	[81]
Male C57BL/6, C3H/HouJ and TLR4 mutant C3H/HeJ mice	Animal	Methionine/choline-deficient diet and weekly intravenous injections of clodronate liposomes	(↑): Steatohepatitis histological condition, portal endotoxemia and TLR4 expression in control mice fed MCDD. (↓): Liver injury and lipid accumulation marker in TLR4 mutant mice. Intravenous injections of clodronate liposomes: depleting liver Kupffer cells → changes in histological condition of steatohepatitis and prevented increases in TLR4 expression.	[72]
TLR4 mutant C3H/HeJ mice and wildtype C3H/HouJ mice	Animal	Water enriched with 30% fructose	(↑): Hepatic steatosis and plasma ALT levels in wildtype mice fed fructose. (↓): Hepatic triglyceride accumulation in TLR4 mutant mice fed fructose. Hepatic lipid peroxidation, MyD88, and TNF-α levels were significantly decreased in TLR4 mutant mice fed fructose group in comparison to wildtype mice fed fructose.	[74]
Inflammasome-deficient mice and *Asc* and *Il18*-deficient mice	Animal	NASH model: methionine-choline-deficient diet for 24 days High fat diet model: 60% calories from fat for 10–12 weeks	(↑): Severity of NASH in inflammasome-deficient mice, *Asc* and *Il18*-deficient mice. Co-housing of inflammasome-deficient animals to wild type mice: exacerbation of hepatic steatosis and metabolic dysfunctions, alteration of gut microbiota configuration.	[77]
Obese patients (*n* = 52)	Human		(↑): Expression of mRNA of TNF-α and TNF receptors p55 in hepatic tissue and peripheral fat of patients with NASH.	[79]

(↑), an increase in condition; (↓), a decrease in condition; ALT, Alanine transaminase; *Asc*, Apoptosis-associated speck-like protein containing a C-terminal caspase recruitment domain; *Il18*, Interleukin18; MCDD, Methionine/choline-deficient diet; MyD88, Myeloid differentiation factor 88; NASH, Non-alcoholic steatohepatitis; TLR4, Toll-like receptor 4; and TNF-α, Tumor necrosis factor-α.

**Table 2 ijms-22-08309-t002:** Alcoholic liver disease studies related to the immune response.

Species	Study Types	Exposure	Main Results	Ref.
Germ-free NIH Swiss female mice	Animal	Oral gavage with alcohol (5 mg/kg)	(↓): Alcohol-induced liver injury, neutrophil infiltration, and levels of pro-inflammatory cytokines were lower in the germ-free mice group than in the other alcohol-fed mice groups. Gut microbiota plays a key role in liver injury through alcohol-induced dysbiosis	[111]
Germ-free C57BL/6 mice	Animal	Oral gavage with acute alcohol (3 g/kg)	(↑): The absence of gut microbiome increases alcohol susceptibility to binge drinking and increases ethanol metabolism in the liver. Acute alcohol supply increased liver inflammation in the sterile mice group due to binge-induced liver damage. In acute alcoholic liver disease, the gut microbiota may play a protective role in inflammation and hepatic steatosis.	[112]
Male Wistar rats	Animal	Continuous ethanol supply for 3 weeks. gut sterilization with polymyxin B and neomycin	(↓): Plasma endotoxin levels (80–90 pg/mL → <25 pg/mL), average hepatic pathological score in ethanol-fed and antibiotic-treated rats Antibiotic treatment prevented elevated aspartate aminotransferase levels and hepatic surface hypoxia.	[99]
Alcohol-fed NS5A Tg mice	Animal	Lieber–DeCarli diet containing 3.5% ethanol or isocaloric dextrin for long-term alcohol feeding, repetitive LPS injection	(↑): Ethanol-induced endotoxemia, liver injury and tumorigenesis after TLR4 induction through hepatocyte-specific transgenic expression of the HCV nonstructural protein NS5A.	[106]
Male C57BL/6J mice	Animal	Administered epinephrine for 5 days (2 mg/kg per day) or bolus ethanol for 3 days (6 g/kg per day), 24 h later, inject LPS (10 mg/kg)	(↑): Severity of liver damage and inflammation due to LPS through prior exposure to epinephrine and ethanol. (↓): Sensitivity of ethanol to liver damage due to co-administration of ethanol and propranolol. Sympathetic nerves influence the progression of ALD.	[107]
Male Wistar rats	Animal	Chronic ethanol diet fed	(↑): ROS production by LPS in Kupffer cells isolated from ethanol-fed mice. ROS production in Kupffer cells by LPS stimulation is increased NADPH oxidase-dependently. ERK1/2 contributes to the increase of TNF-α production in Kupffer cells by LPS stimulation.	[108]
Patients (*n* = 14: alcoholic hepatitis 8, cirrhotic with alcoholic hepatitis 5, severe alcoholic hepatitis 1)	Human		(↑): Plasma endotoxin levels and Serum IL-6 and IL-8 levels of patients compared to healthy subjects. Serum LBP was positively correlated with white blood cell and neutrophil counts as an indicator of an inflammatory response.	[100]
Controls (*n* = 11), Alcoholics (*n* = 30: minimal patients: 10, intermediate patients: 9, cirrhotic alcoholic liver disease patients: 11)	Human		(↑): Endotoxin levels and endotoxin activity-related binding factors concentration in alcoholic groups	[101]
Controls (*n* = 6), patients with alcoholic hepatitis (*n* = 6)	Human		(↑): nuclear factor-κB activity in the monocytes of 6 patients with alcoholic hepatitis as compared with normal subjects. (↑): Nuclear factor-kB activity, TNF-α RNA expression and TNF-α release by endotoxin in alcoholic hepatitis patients.	[109]
Controls (*n* = 11), patients with alcohol use disorder (*n* = 42) and alcoholic hepatitis (*n* = 91)	Human		(↑): Retention levels of ECE1 in individuals according to alcoholic patient severity Genetically engineered *C. albicans* strain exacerbates ethanol-induced liver disease in mice and increases mortality in mice. Candidalysin can exacerbate ethanol-induced liver disease and damage hepatocytes independently of the β-glucan receptor.	[110]

(↑), an increase in condition; (↓), a decrease in condition; ALD, Alcoholic liver disease; ECE1, extent of cell elongation 1; ERK1/2, extracellular signal-regulated protein kinase; HCV, Hepatitis C virus; IL, interleukin; LBP, Lipopolysaccharide binding protein; LPS, Lipopolysaccharide; NADPH, Nicotinamide adenine dinucleotide phosphate; ROS, reactive oxygen species; TLR4, Toll-like receptor 4; and TNF-α, Tumor necrosis factor-α.

**Table 3 ijms-22-08309-t003:** Liver cirrhosis studies related to the immune response.

Species	Study Type	Exposure	Main Results	Ref.
Germ-free C57BL/6 male mice	Animal	TAA or CCl_4_	(↑): Liver fibrosis was increased in the germ-free mice group compared to the control mice. More toxin-induced oxidative stress and cell death were observed. The commensal gut microbiota prevents liver fibrosis in conditions of chronic liver injury.	[127]
Male Sprague-Dawley rats	Animal	Administration of CCl_4_ and fed phenobarbital in drinking water (35 mg/dL)	Bacterial translocation was seen in 48% of cirrhosis rat models. Cirrhosis rat model with small intestinal bacterial overgrowth had a significantly higher bacterial translocation rate and slower intestinal transit rate compared to the control group.	[122]
Male Sprague–Dawley rats	Animal	Subcutaneous injection of an equal mixture of CCl_4_ and olive oil. antibiotic (norfloxacin) and different probiotic treatments	(↑): Levels of *Enterobacteriaceae* compared to controls in a cirrhosis rat model. (↑): Levels of *Lactobacillus* in the cirrhotic rat group treated with *Bifidobacteria* compared to the saline treated group. (↓): Levels of *Enterobacteriaceae* in the cirrhotic rat group treated with *Bifidobacteria* compared to the saline treated group. (↓): Levels of endotoxin in the cirrhotic rat group respectively treated with *Bifidobacteria* and *Enterococcus* compared to the saline treated group.	[123]
Male C3H/HeOuJ mice (TLR4 wild type), C3H/HeJ mice (TLR4 mutant), *Tlr2* deficient mice, *Trif*^Lps2/Lps2^ mice, C57BL/6 mice and MyD88 deficient mice	Animal	Underwent bile duct ligation. fed CCl4 or TAA	TLR4 and the gut microbiota play an essential role in liver fibrogenesis. (↑): TGFβ-mediated activation of hepatic stellate cells and collagen production. (↓): Regulation of TGFβ pseudo-receptor Bambi in quiescent hepatic stellate cells.	[126]
Controls (*n* = 45), Patients (*n* = 169)	Human		(↑): Plasma endotoxin levels of chronic hepatitis patients and cirrhosis patients compared with healthy subjects. Endotoxemia was identified in chronic hepatitis patients (27%), chronic hepatitis patients with acute exacerbation (85%) and cirrhosis patients (41%), respectively. In cirrhosis patients, plasma endotoxin levels increased progressively in association with the severity of liver dysfunction.	[119]
Non-infected cirrhosis patients (*n* = 75: 55 ascites and 20 no ascites)	Human		Bacterial DNA detection only in patients with ascites. Presence of bacterial DNA in plasma contributed to systemic hemodynamic impairment in patients with ascites cirrhosis and exacerbated intrahepatic endothelial dysfunction in cirrhosis.	[120]
Cirrhosis cohort patients (*n* = 53)	Human		Small intestinal bacterial overgrowth was seen in 59% of patients with cirrhosis and was significantly related to systemic endotoxemia.	[121]

(↑), an increase in condition; (↓), a decrease in condition; CCl_4_, carbon tetrachloride; MyD88, Myeloid differentiation factor 88; TAA, thioacetamide; TGFβ, transforming growth factorβ; and TLR, Toll-like receptor.

## Data Availability

Data is contained within the article.

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
