# Peer review of "The Gut Microbiota-Derived Immune Response in Chronic Liver Disease"

_ijms, 2021, doi:10.3390/ijms22158309_

Round 1
Reviewer 1 Report
Focus of review is relevant and timely however breadth and depth of discussion is lacking.
Clarification is needed why Authors have limited scope of review to ALD and NASH given wealth of data supporting role of gut-liver axis in many other chronic liver diseases (PBC, PSC, AIH, etc).
Better integration of data in tables with text, particularly examples with animal models is needed - these findings seem randomly inserted into tables but models are not placed in context of their usefulness. What about germ-free animal experiments? What have these shown?
Very little is mentioned regarding non-TLR mediators? What about metabolites? What about other factors?
Overall concepts should be better highlighted - what are the overall implications of current findings? Where are the knowledge gaps (there are many), how might we address these unknowns and how might this influence our understanding of disease mechanisms and inform possible treatments?
Author Response
ijms-1303810
“The gut microbiota-derived immune response in chronic liver disease”
Point-to-point responses to comments by the Reviewer 1
First of all, we would like to thank the Reviewer 1 for his/her comments, which helped us to improve this manuscript.
Specific Comments:
- Comment 1: Clarification is needed why Authors have limited scope of review to ALD and NASH given wealth of data supporting role of gut-liver axis in many other chronic liver diseases (PBC, PSC, AIH, etc).
- Response 1: We set the scope by dividing it into NAFLD and ALD in a large group, but we modified it by adding "Other Liver Disease" part to the need for explanation and discussion of additional diseases. The following paragraph was added to the section.
“5.4. Hepatocellular Carcinoma and the Immune Response
The pathogenesis of HCC is caused by a combination of various factors. Non-viral HCC may contribute to the pathogenesis of the disease through hepatic steatosis, oxida-tive stress, and endoplasmic reticulum stress, and dysbiosis and resulting inflammation are attracting attention as an additional factor. Data from clinical and animal studies allow us to observe extreme changes in the composition of the gut microbiome in host with HCC. In clinical studies with HCC patients, an excessive increase in E. coli was confirmed. In addition, animal models and human studies have confirmed the presence of Helicobacter spp. in liver tissue samples. It has been suggested as a po-tential mechanism by which Helicobacter migrates to HCC tumor tissue via intestinal translocation to suppress anti-tumor immunity and cause activation of NF-kB signaling to promote carcinogenesis. But strangely, Helicobacter was not found in patients with viral HCC.
Dysbiosis, which is associated with HCC, increases bacterial translocation and circulation of carcinogens through disruption of the intestinal barrier and contributes to the activation of several proinflammatory and oncogenic signaling pathways. Additionally, the role of the gut microbiome on liver tumorigenesis has been clearly identified in animal studies. Attenuation of liver inflammation and HCC development was observed using wide-spectrum antibiotics in mice model. In a mouse model in which HCC was induced through carcinogens, the influx of LPS led to activation of TLR4, followed by activation of NF-kB pathway in HSC, resulting in increased tumor cell proliferation.
As the understanding of the gut microbiome increases, various therapeutic approaches for HCC are being devised. Research on reducing the risk of developing HCC in NAFLD through synthetic bile acids, and suppressing tumor size and growth by administering probiotics to HCC mouse models is being actively conducted. However, further clinical studies are needed to link and characterize the role of the gut microbiota with the pathogenesis of HCC.
5.3. Other Liver Disease and the Immune Response
Hepatic encephalopathy is a cognitive impairment caused by serious liver disease, and it has been confirmed that there is a deep relationship with the gut microbiome and metabolites. Ammonia is a key factor in the onset of hepatic encephalopathy, and over-growth of urease-activated bacteria is known to be the main cause of hyperammonemia. Recently, however, the focus of ammonia-generating sources has changed from the large intestine to the small intestine and kidney. Other factors that aggravate and develop hepatic encephalopathy include systemic inflammation and endotoxemia caused by dysbiosis, and animal studies have confirmed that these factors cause aggravation of coma and cytotoxic edema in cirrhosis model. In addition, a decrease in neuro-psychological function has been confirmed after hyperammonemia is induced in cirrhosis patients with inflammation and infection. As a therapeutic approach for hepatic encephalopathy, modulation of the gut microbiota is attracting attention as an alternative, and the effects of prebiotics, probiotics and symbiotics are being verified. In addition, a therapeutic approach through the administration of rifaximin has been confirmed to be safe and to efficiently control the gut microbiome, and many studies are being conducted.
Primary sclerosing cholangitis (PSC) is a chronic liver disease characterized by biliary inflammation and stenosis of the bile ducts. Various animal studies over the past few years have provided supporting data for a causal relationship between the gut microbiota and PSCs. Dysbiosis, bacterial translocation by weakening the intestinal barrier, and the immune response associated with the pathogenesis of PSC are considered key factors. In animal studies, transplantation of fecal microbiota from PSC patients into germ-free mice showed that the PSC phenotype was transferred and that increased susceptibility to hepatobiliary damage by diethyldithiocarbamate. In addition, in the MDR2-/- mouse model mimicking human PSC, we were able to confirm dysbiosis and increased intestinal permeability. The gut microbiota contributes to PSC pathogenesis through interventions in the synthesis and production of various metabolites, including bile acids, which influence disease pathogenesis as signaling molecules in the gut-liver axis. Although the identification of a potential causal relationship be-tween the gut microbiota and PSCs in animal studies has made significant progress, validation and reproducibility in clinical studies is still minimal.
Primary biliary cholangitis (PBC), previously known as primary biliary cirrhosis, is a chronic liver disease caused by immune cell activation and damage to the bile ducts. PBC, which had been considered a typical autoimmune disease, through gut-liver axis interactions, has led to the inclusion of the influence of the gut microbiota in a revised etiological understanding. Microbial-associated molecular patterns by dysbiosis can lead to persistent inflammation in the bile duct and aggravate the disease. The presence of pyruvate dehydrogenase complex E2 subunit antimitochondrial antibodies (AMA) is a representative serological characteristic of PBC, which cross-reacts with proteins of strains such as Escherichia coli and Novosphingobium aromaticivorans, resulting in immune attack on biliary epithelial cells. In animal studies, PBC-mimicking liver disease occurred in mice when a specific mouse model was infected with Novosphingobium aromaticivoransm. This is meant as supporting evidence for the association of PBC with the gut microbiome. Novel therapeutic approaches based on the microbiome appear to be necessary for PBC management, and additional clinical data and clinical studies are needed.”
- Comment 2: Better integration of data in tables with text, particularly examples with animal models is needed - these findings seem randomly inserted into tables but models are not placed in context of their usefulness. What about germ-free animal experiments? What have these shown?
- Response 2: The study cases of germ-free mice were investigated and additionally applied to the cases in the table. We also tabulated and added cases for diseases other than NAFLD, ALD, and cirrhosis.
- Comment 3: Very little is mentioned regarding non-TLR mediators? What about metabolites? What about other factors?
- Response 3: We have identified the need to address other factors that affect disease. We added mention of bile acids, choline metabolites and ethanol products.
“Bile acids (BAs) are synthesized from cholesterol in hepatocytes, bound to glycine or taurine, and released into the bile ducts. Promotes the emulsification and absorption of fats, cholesterol, and fat-soluble vitamins in the small intestine, and 95% of the bile acids are reabsorbed back to the liver in ileum. The remaining 5% is reprocessed by the gut microbiota and reaches the liver through the portal vein in the form of secondary bile acids. This enterohepatic circulatory system plays an important role in maintaining homeostasis as part of the gut-liver axis. BA directly controls the gut microbiota and binds with FXR to induce the production of antimicrobial peptides such as angogenin1. Through this, it suppresses intestinal microbial overgrowth and intestinal barrier dysfunction. However, dysbiosis disrupts primary and secondary bile acid circulation and enterohepatic circulation balance and triggers a series of host immune responses, contributing to the progression of liver disease’
“In NAFLD, choline metabolites are one of the important factors in the pathogenesis and progression of the disease. Choline is an essential nutrient important for maintaining a healthy metabolism and plays a key role in liver function and brain development and nerve function. Choline plays a role in helping the liver to excrete particles of very-low density lipoproteins and prevents hepatic steatosis. These properties allow a choline-deficient diet to mimic nonalcoholic steatohepatitis in a mouse model. Choline is converted to trimethylamine (TMA) by the intestinal microflora and then to trimethylamine N-oxide (TMAO), which can be transported to the liver. Increased systemic circulation of TMAO leads to hepatic steatosis, which leads to liver damage and is another cause of NAFLD”
“Ethanol and its metabolites acetaldehyde and acetate are closely linked to liver damage. It also damages the intestinal barrier, allowing for bacterial translocation and entry of microbial products and causes weakening of tight junctions. Ethanol down-regulates the expression of antimicrobial peptides in the intestine, weakening the inhibition of bacterial overgrowth, which also leads to a decrease in intestinal butyrate”
- Comment 4: Overall concepts should be better highlighted - what are the overall implications of current findings? Where are the knowledge gaps (there are many), how might we address these unknowns and how might this influence our understanding of disease mechanisms and inform possible treatments?
- Response 4: We added additional information and supplementary content. We corrected figure contents and added paragraphs to the section for each disease. We have added paragraphs as follows.
“Imbalances and dysregulation of the immune system in the gut and liver are associated with the onset and progression of intestinal and liver disease. While the mucosal surface of the intestinal barrier serves as a primary barrier, mucus protects the basal epithelium, induces immunomodulatory signals, and maintains and enhances homeostasis. In the porous mucosal layer present in the small intestine, MUC2 mucin is directly absorbed by dendritic cells, imprinting anti-inflammatory properties in the dendritic cells. These actions inhibit the expression of inflammatory cytokines by inhibiting gene signaling through nuclear factor-κB. Induction of regulatory signals in these dendritic cells of MUC2 limits and modulates the immunogenicity of gut antigens. The dendritic cells then migrate to the mesenteric lymph nodes and present antigens that stimulate Treg cells and effector T cells. These cells deliver regulatory cytokines such as TGF-β, IL-10, and IL-35 throughout the body and carry out immune responses, while safeguarding the balance of gut and immunity.”
“Microbial-associated molecule patterns, such as LPS, lipoteichoic acid, peptidoglycan and lipoproteins that are released into the portal vein, are detected by immune cells expressing pattern recognition receptors and trigger an activation”
“In summary, intestinal barrier dysfunction and translocation of bacteria and thus products through them can be a significant cause of chronic liver disease and related complications”
“The 'leaky gut' hypothesis still links microbial products in the gut with the pathogenesis and progression of NAFLD and ALD and has long been considered one of the major contributors. Compared with healthy control, patients with NAFLD show increased intestinal permeability and tight junctions, and chronic alcohol abuse contributes to disruption of the intestinal barrier, which is critical for the development and progression of ALD, sup-porting this hypothesis.”
“However, what makes NAFLD so distinct from obesity is the difference in a process called lipotoxicity. In the process of liver lipid overload, the way liver cells deal with it is either steatosis adaptation or induction of cell death by molecular mechanisms. Stress signals released from hepatocytes due to cell death trigger activation of inflammatory pathways and, over time, lead to abnormal wound repair processes such as chronic injury and liver fibrosis. In this way, NAFLD can deepen and progress to NASH and fibrosis. Recently, the influence of the gut microbiome according to the gut-liver axis relationship as a factor in the novel NAFLD pathogenesis has been attracting attention.”
“Chronic inflammation is a key etiology of ALD, and it has been reported that ethanol itself enhances the ability of immune cells to respond to inflammatory stimuli. Alcohol expo-sure upregulates the expression of TLRs and enhances the signaling of NF-kB and Early Growth Response-1, induced proinflammatory cytokine production. Additionally, it increases the proinflammatory activity of hepatic NKT cells and promotes the pro-duction of chemokines such as IL-8 and MCP-1”
Reviewer 2 Report
Authors mentioned the association of gut-microbiota derived immune response with also HCC in the introduction section. But no remark about HCC was found. Authors should also describe that of HCC. Also hepatic encephalopathy should also be mentioned.
Author Response
ijms-1303810
“The gut microbiota-derived immune response in chronic liver disease”
Point-to-point responses to comments by the Reviewer 2
First of all, we would like to thank the Reviewer 2 for his/her comments, which helped us to improve this manuscript.
Specific Comments:
- Comment 1: Authors mentioned the association of gut-microbiota derived immune response with also HCC in the introduction section. But no remark about HCC was found. Authors should also describe that of HCC. Also hepatic encephalopathy should also be mentioned.
- Response 1: We have added the discussion related to HCC and hepatic encephalopathy that reviewer mentioned.
5.4. Hepatocellular Carcinoma and the Immune Response
The pathogenesis of HCC is caused by a combination of various factors. Non-viral HCC may contribute to the pathogenesis of the disease through hepatic steatosis, oxidative stress, and endoplasmic reticulum stress, and dysbiosis and resulting inflammation are attracting attention as an additional factor [128]. Data from clinical and animal studies allow us to observe extreme changes in the composition of the gut microbiome in host with HCC. In clinical studies with HCC patients, an excessive increase in E. coli was confirmed [129]. In addition, animal models and human studies have confirmed the presence of Helicobacter spp. in liver tissue samples [130]. It has been suggested as a potential mechanism by which Helicobacter migrates to HCC tumor tissue via intestinal translocation to suppress anti-tumor immunity and cause activation of NF-kB signaling to promote carcinogenesis [131]. But strangely, Helicobacter was not found in patients with viral HCC.
Dysbiosis, which is associated with HCC, increases bacterial translocation and circulation of carcinogens through disruption of the intestinal barrier and contributes to the activation of several proinflammatory and oncogenic signaling pathways [132]. Additionally, the role of the gut microbiome on liver tumorigenesis has been clearly identified in animal studies. Attenuation of liver inflammation and HCC development was observed using wide-spectrum antibiotics in mice model. In a mouse model in which HCC was induced through carcinogens, the influx of LPS led to activation of TLR4, followed by activation of NF-kB pathway in HSC, resulting in increased tumor cell proliferation [133].
As the understanding of the gut microbiome increases, various therapeutic approaches for HCC are being devised. Research on reducing the risk of developing HCC in NAFLD through synthetic bile acids [134], and suppressing tumor size and growth by administering probiotics to HCC mouse models is being actively conducted. However, further clinical studies are needed to link and characterize the role of the gut microbiota with the pathogenesis of HCC.
Round 2
Reviewer 1 Report
Thank you for added changes. Could you please upload a PDF of the revised manuscript (v.2) that includes the new text in a different font colour. It is difficult to know where some of the revisions have been added to v.2.
Author Response
ijms-1303810
“The gut microbiota-derived immune response in chronic liver disease”
Point-to-point responses to comments by the Reviewer 1
First of all, we would like to thank the Reviewer 1 for his/her comments, which helped us to improve this manuscript.
We added v2 (first revision) PDF file
In case of v3 (second revision) PDF, we added PDF file on the supplementary file section on submission site. we can not submit file on this site because only one file can be uploaded.
Sincerely
Ki Tae Suk, M.D., Ph.D.
Department of Internal Medicine, Division of Gastroenterology and Hepatology, Hallym University College of Medicine, Chuncheon, 24253, South Korea.
ktsuk@hallym.ac.kr
Telephone: +82-33-240-5826, Fax: +82-33-241-8064

Reviewer 2 Report
Revised manuscript was well-written and informative for clinicians. The association between liver diseases and gut microbiota is an important theme, and therefore this review is useful for readers to understand it comprehensively.
Author Response
ijms-1303810
“The gut microbiota-derived immune response in chronic liver disease”
Point-to-point responses to comments by the Reviewer 2
First of all, we would like to thank the Reviewer 2 for his/her comments, which helped us to improve this manuscript.
Sincerely Yours,
Ki Tae Suk, M.D., Ph.D.
Department of Internal Medicine, Division of Gastroenterology and Hepatology, Hallym University College of Medicine, Chuncheon, 24253, South Korea.
ktsuk@hallym.ac.kr
Telephone: +82-33-240-5826, Fax: +82-33-241-8064
Round 3
Reviewer 1 Report
Submitted revisions substantially elevate the manuscript quality.
Several key references/concepts remain lacking from Review and should be discussed briefly for each liver disease (if findings are available). Examples below:
- translational influence of dysbiosis on gut metabolism; see PMID: 33387530
- identification of pathogenic bugs and appreciation of interconnected networks leading to liver inflammation/disease; see PMID: 30643240
- potential treatment approaches for dysbiosis in liver disease; see PMID: 31723265
Minor points
line 150 - unclear sentence structure
Tables - placement of 'Animal' or 'Human' labels within Tables is inconsistent and therefore unclear. Authors should include a 'species' column as 1st column in Table and/or centre 'Animal' and 'Human' categorization within the table field (i.e. the 'Animal' and 'Human' labels for each table appear randomly beside certain examples). Overall format of information in Tables could be tidied/simplified to ease readability (e.x. Table 2, 'Conditions', alcoholic hepatitis (n=8), cirrhotic alcoholic hepatitis (n=5), etc. rather than 'fourteen patients......'
Subheading 5.3 appears incorrect. Should it not be 5.5?
Author Response
ijms-1303810
“The gut microbiota-derived immune response in chronic liver disease”
Point-to-point responses to comments by the Reviewer 1
First of all, we would like to thank the Reviewer 1 for his/her comments, which helped us to improve this manuscript.
Specific Comments:
- Comment 1: Several key references/concepts remain lacking from Review and should be discussed briefly for each liver disease (if findings are available). Examples below: translational influence of dysbiosis on gut metabolism; see PMID: 33387530, identification of pathogenic bugs and appreciation of interconnected networks leading to liver inflammation/disease; see PMID: 30643240, potential treatment approaches for dysbiosis in liver disease; see PMID: 31723265
- Response 1: We appreciate the Reviewer’s thoughtful comment. We discussed briefly for each liver disease.
“Another study suggested that patients with alcoholic hepatitis have increased fecal numbers of Enterococcus faecalis and bacteriophages can specifically target cytolytic E. faecalis, which provides a method for precisely editing the intestinal microbiota. (Nature. 2019 Nov;575(7783):505-511.)”
“In another study with 2 cohorts, the gut microbiome of PSC patients reveals functional differences compared with that of control group, including microbial metabolism of essential nutrients.(Gastroenterology. 2021 Apr;160(5):1784-1798.e0.) Regarding epithelial barrier dysfunction, K. pneumoniae disrupts the epithelial barrier to initiate bacterial trans-location and liver inflammatory responses in PSCs patients. In this study, antibiotic treatment ameliorated the T helper 17 immune response induced by PSC-derived micro-biota (Nat Microbiol. 2019 Mar;4(3):492-503. doi: 10.1038/s41564-018-0333-1.)”
- Comment 2: line 150 - unclear sentence structure
- Response 2: Thanks for these suggestions. It has been rewritten by correcting the unclear part of the subject of the sentence. It has been modified as follows.
“BAs promotes the emulsification and absorption of fats, cholesterol, and fat-soluble vitamins in the small intestine, after which 95% of the BAs are reabsorbed from the ileum back to the liver”
- Comment 3: Tables - placement of 'Animal' or 'Human' labels within Tables is inconsistent and therefore unclear. Authors should include a 'species' column as 1st column in Table and/or centre 'Animal' and 'Human' categorization within the table field (i.e. the 'Animal' and 'Human' labels for each table appear randomly beside certain examples). Overall format of information in Tables could be tidied/simplified to ease readability (e.x. Table 2, 'Conditions', alcoholic hepatitis (n=8), cirrhotic alcoholic hepatitis (n=5), etc. rather than 'fourteen patients......'
- Response 3: We agree with the reviewer’s comment and apologize for causing confusion. We corrected the labeling of 'animal' or 'human'. As reviewer said, I moved the label to the middle and reclassified it. The number notation in 'Human' was unified and changed to (n = ....).
- Comment 4: Subheading 5.3 appears incorrect. Should it not be 5.5?
- Response 4: The part that was incorrectly marked as 5.3 was corrected to 5.5.
